# Early Detection Methods for Toxic Cyanobacteria Blooms

**DOI:** 10.3390/pathogens13121047

**Published:** 2024-11-28

**Authors:** Lauren Grant, Diane Botelho, Attiq Rehman

**Affiliations:** 1Department of Chemistry, Saint Mary’s University, 923 Robie Street, Halifax, NS B3H 3C3, Canada; lo.grant@smu.ca; 2New Brunswick Research and Productivity Council (RPC), 921 College Hill Rd, Fredericton, NB E3B 6Z9, Canada; diane.botelho@rpc.ca

**Keywords:** cyanobacteria, cyanotoxins, harmful algal blooms, ELISA, LCMS, qPCR

## Abstract

Harmful cyanobacterial blooms produce cyanotoxins which can adversely affect humans and animals. Without proper monitoring and detection programs, tragedies such as the loss of pets or worse are possible. Multiple factors including rising temperatures and human influence contribute to the increased likelihood of harmful cyanobacteria blooms. Current approaches to monitoring cyanobacteria and their toxins include microscopic methods, immunoassays, liquid chromatography coupled with mass spectrometry (LCMS), molecular methods such as qPCR, satellite monitoring, and, more recently, machine learning models. This review highlights current research into early detection methods for harmful cyanobacterial blooms and the pros and cons of these methods.

## 1. Introduction

Cyanobacteria blooms can adversely affect human and animal health through the production of various cyanotoxins. Some of the most common cyanotoxins are hepatotoxins and neurotoxins [1]. In 2018, three dogs died after ingesting anatoxins in benthic mats from the Walostoq River in New Brunswick, Canada. The dog owners said there was no water discoloration, indicating the toxic cyanobacteria were not surface-blooming species [2]. Similarly, two dogs died from anatoxin poisoning after ingesting benthic cyanobacteria mats on the shore of Shubenacadie Grand Lake in Nova Scotia, Canada, in June 2021 [3].

To prevent future tragic incidents like those in New Brunswick and Nova Scotia, establishing robust protocols for assessing cyanobacteria in water is crucial. Health Canada’s guideline for total microcystins in recreational water is 10 µg/L and 1.5 µg/L in drinking water [4,5]. The United States Environmental Protection Agency (EPA) health advisories for microcystins and cylindrospermopsin in drinking water for school-age children and adults is 1.6 µg/L and 3.0 µg/L, respectively. For infants and pre-school-aged children, the health advisory is 0.3 µg/L for microcystins and 0.7 µg/L for cylindrospermopsin [6]. Any methods employed for cyanotoxin detection should have a limit of detection (LOD) at or below these health advisory levels.

Cyanobacteria blooms are more likely to occur in warmer water above 25 °C with abundant nutrients such as nitrogen, phosphorus, and carbon [7]. Excessive use of fertilizers leading to runoff contributes to nutrient availability and encourages waters’ eutrophication. Sewage runoff can also add to this, and, combined with the impacts of climate change, these factors contribute to the rising occurrence of harmful cyanobacteria blooms [8]. In a study of Canadian lakes in Alberta, Saskatchewan, and Manitoba, Erratt et al. [9] found that greater human influence led to earlier breakpoints for increased cyanobacterial biomass. They also found that rising temperatures encouraged the increase in microcystin production potential in lakes.

With rising temperatures and other factors that increase toxic cyanobacteria bloom potential, it is important to have effective monitoring programs in place. Methods for detecting toxic cyanobacteria include traditional microscopic methods, immunoassays, chromatography and mass spectrometry, PCR-based methods, satellite imaging, and machine learning models. Although each method has pros and cons, this review specifically focuses on their applicability to early warning systems.

## 2. Materials and Methods

A comprehensive search of the academic databases PubMed, Web of Science, Google Scholar, and Novanet, an online consortium of 11 university libraries in Nova Scotia and New Brunswick, was conducted. Specific terms, such as “Cyanobacteria and their toxins”, “Harmful algal blooms (HABs)”, “Cyanotoxins and Early Detection”, “Cyanotoxins and sampling”, and “Cyanobacteria/Cyanotoxin detection methods”, were searched. The titles and abstracts of the articles were screened and assessed for inclusion in the review. A total of 59 papers were more thoroughly examined to draw conclusions on the current state of early detection methods for harmful cyanobacteria blooms in the literature. Finally, 49 references were included in the present paper, of which 60% were published from 2020 onward.

## 3. Cyanotoxin Detection Methods

### 3.1. ELISA

Enzyme-linked immunosorbent assay (ELISA) is a biochemical assay that detects the presence of an antigen. An enzyme-labeled antibody, which specifically binds to the antigen, is added, and any unbound antibodies are then washed away. A chromogenic substrate is then added to react with the enzyme and a change in colour or fluorescence can be measured to identify the presence of the antigen [10].

ELISA results can be impacted by non-specific binding and cross-reactivity [11]. Most of the microcystin antibodies for ELISA have been made from microcystin-LR, which can impact the selectivity of the antibody for other microcystins [11]. This issue can be mitigated by using the conserved “Adda” amino group in all microcystins and nodularins [12]. However, the Adda antibody selectivity can still be impacted in environmental samples where some Adda groups may have been modified, for example, to dimethyl Adda or acetylate Adda [11]. However, ELISA is a well-developed assay with low costs and a high throughput [13].

Liu et al. [13] looked at the effects of sample pre-treatment on the quantitative detection of cyanotoxins by ELISA. Liu et al. [13] performed two pre-treatment methods and found that initial centrifugation and the filtering of surface water samples do not capture intracellular microcystins. The alternative method utilized a freeze–thaw pre-treatment to lyse cells, allowing for the quantification of both extracellular and intracellular microcystins. The impact of dilution on sample matrices tested (tap, river, and lake water) was also investigated. Liu et al. [13] found the optimal dilution factor for reducing matrix effects was 2:1 for tap water and 4:1 for lake and river water with an anti-matrix buffer containing 10× phosphate buffer solution (PBS), 1% bovine serum albumin (BSA), and 0.5% ethylene diamine tetraacetic acid (EDTA). The limit of detection (LOD) was 0.15 µg/L and the limits of quantification were between 0.27 µg/L and 1.87 µg/L for microcystin-LR.

Woodruff et al. [14] also employed ELISA for the detection of microcystins, but with a focus on water facility sludges. Cyanobacteria caught in sludges have been found to increase in numbers and release toxins into the solution. Woodruff et al. [14] specifically looked at the impact of sample storage conditions, temperature, and holding time on microcystin-LR recovery efficiency using ELISA compared with LC-MS/MS. Woodruff et al. [14] found that acceptable recovery of microcystins after storage at 4 °C for 24 h and −20 °C for 7 and 21 days was achieved. They highlighted that this adds to the accessibility of ELISA as samples can be stored inexpensively, helping to minimize costs. A linear response was also found for spiked sludge samples with varying total organic carbon and turbidity. The authors conclude that although their results are promising for monitoring water facility sludges for microcystins using ELISA, LC-MS/MS is still needed in cases requiring better accuracy and precision.

ELISA has also been used for cyanotoxin identification in benthic mats. Bauer et al. (2023) [15] used ELISA to determine saxitoxin concentrations in benthic mats from a river in Lech, Germany, where previously three dogs were poisoned by benthic cyanobacteria. The highest concentration of saxitoxin found was 0.59 µg/L in benthic mats and 0.35 µg/L in open water samples. The WHO has a guideline value of 3 µg/L for saxitoxins in drinking water [16]. In another study, Bauer et al. (2022) [17] used microscopic methods to determine if the benthic cyanobacteria, *Tychonemas* sp., was present in samples. Anatoxin-a concentrations in 50 mL samples containing water, sediment, and biomass were determined using ELISA. The highest concentration of anatoxin-a was 220.5 ng/L, found in a sample from a bathing site along the river.

Unlike microcystins and cylindrospermopsin, anatoxin-a has less well-established guidelines for warning levels in recreational and drinking water. Microcystins have fairly consistent guideline values; for example, Health Canada suggests 1.5 µg/L and the EPA suggests 1.6 µg/L [4,6]. However, Table 1 outlines some of the guidance values which have been issued for anatoxin-a. The biggest discrepancy in these guidelines comes from the WHO. However, in the report from the WHO, it was noted that there is insufficient information to develop long-term health-based reference values for anatoxins [18]. The “Guidelines for Canadian drinking water: guideline technical document—cyanobacterial toxins” published by Health Canada also states there is insufficient evidence to develop a maximum acceptable concentration in drinking water for anatoxins [4]. However, Bauer et al. (2022) [17] were able to measure anatoxin levels using ELISA at levels below more conservative estimates for anatoxin guidance values in drinking water.

Cyanotoxins other than microcystins and anatoxins have also been measured using ELISA. Roy-Lachapelle et al. [21] evaluated an ELISA kit for the determination of anabaenopeptins, a cyanopeptide of emerging concern. Anabaenopeptins are protease inhibitors with IC_50_ values at a nanomolar scale. They have been found in concentrations similar to microcystins in lakes in the United States and Europe [22]. Roy-Lachapelle et al. [21] tested the ELISA total anabaenopeptin kit against a SPE-UHPLC-HRMS method for quantifying anabaenopeptin A and B. They found that the ELISA test was reproducible when proper blanks were used. However, the LCMS method had greater sensitivity and a limit of detection of 0.011 µg/L compared with 0.10 µg/L for ELISA.

### 3.2. Liquid Chromatography Mass Spectrometry

Chromatographic methods can have a higher level of accuracy and precision than immunoassays. However, these methods have high costs which makes them poorly suited for regular use in early detection warning systems. Regardless, liquid chromatography–mass spectrometry (LCMS) methods are regularly used for cyanotoxin detection and quantification. LCMS serves as a reliable baseline for comparing the accuracy and precision of other methods [21,23,24,25].

An ultra-trace solid-phase extraction (SPE) ultra-high-performance liquid chromatography high-resolution mass spectrometry (UHPLC-HRMS) method developed by Filatova et al. [26] was able to detect seven microcystins, cylindrospermopsin, anatoxin-a, and nodularin at a picogram per litre scale. Filatova et al. [26] showed good linearity for all cyanotoxins tested and acceptable inter-day and intra-day precision. The SPE-UHPLC-HRMS method was used to test water samples from three reservoirs along the Ter River in Catalonia, Spain. Only one microcystin was found, microcystin-RR, at concentrations between 1 to 2 ng/L. As the authors note, these results support that this method can be used to identify the WHO guideline value for MC-LR in real-world water samples.

Reveillon et al. [27] used solid-phase extraction (SPE) and liquid chromatography tandem mass spectrometry (LC-MS/MS) to look at nine microcystins and one nodularin in cyanobacteria and extracellularly in salt water. This method was developed with estuaries in mind as cyanotoxins can potentially migrate from freshwater to salt waters and impact marine organisms. Two different SPE sorbents were compared, C18 and polymeric. The C18 SPE cartridge was found to have good recovery of extracellular microcystins and nodularins regardless of the presence of salt. This contrasts with the polymeric cartridge which was impacted by salt and was less reliable. For intracellular toxins, cell pellets were lysed using methanol before SPE extraction.

The LC-MS/MS methods used a C18 column with a gradient elution consisting of mobile phase A, water and B, acetonitrile, both with 0.1% formic acid. Reveillon et al. [27] found that the limit of detection (LOD) was between 1.0 and 22 pg and the limit of quantification (LOQ) was between 5.5 and 124 pg for the intracellular matrix. The extracellular matrix had a LOD between 0.59 and 11 pg and a LOQ between 1.8 and 34 pg. Reveillon et al. [27] did not use an internal standard or matrix-matched calibration curve. However, they were able to achieve excellent accuracy using a correction factor. The authors note using a correction factor achieved similar accuracy to methods which used an internal standard or matrix-matched calibration curves. Although, methods using isotopically labelled microcystins as internal standards were even more accurate.

The authors also tested the impacts of increasing salinity on the growth of *Microcystis* spp. and the presence of extracellular cyanotoxins. They found that *Microcystis* spp. could survive increases in salinity while the concentration of extracellular cyanotoxins also increased. This shows that methods for saltwater analysis of cyanotoxins could be crucial for monitoring estuaries which are connected to freshwater bodies containing harmful cyanoblooms.

In a method developed for both fresh and salt water, Vo Duy et al. [28] used an on-line SPE hydrophilic interaction liquid chromatography (HILIC) MS method to quantify saxitoxin, neosaxitoxin, and their decarbamoyl analogues. Five filters were tested for filtration recovery, including glass fibre, nitrocellulose, nylon, polyethersulfone, and regenerated cellulose. All filters except glass fibre provided acceptable recovery of the cyanotoxins tested. The optimal chromatographic conditions using an Accucore 150 Amide HILIC column were 5.0 mM ammonium formate and 0.05% formic acid in HPLC water for mobile phase A and 0.05% formic acid in acetonitrile for mobile phase B.

Vo Duy et al. [28] also evaluated four different methods to mitigate matrix effects. These included an external calibration curve in water, an internal calibration curve in water, an external matrix-matched calibration curve, and an internal matrix-matched calibration curve. Absolute matrix effects were between −62% and −91% for brackish water and between −1.5% and −54% for freshwater using an external calibration curve in water. Absolute matrix effects were greatly improved when matrix-matched isotopically labelled internal standards were used. Brackish water absolute matrix effects were between −11.5% and 1.8% while freshwater absolute matrix effects were between −6.2% and 22.2%, respectively. The LOD ranged from 0.72 to 3.9 ng/L for freshwater samples and 6.9 to 30 ng/L for brackish water. These results highlight the sensitivity of LCMS compared with other methods used for cyanotoxin detection.

### 3.3. Polymerase Chain Reaction (PCR) Methods

Many studies have used PCR-based techniques to determine the presence of cyanotoxin biosynthetic gene clusters in water samples. PCR-based techniques could serve as an invaluable tool for assessing the potential cyanotoxin production in water sources, making it a crucial component of early warning systems. However, important considerations for using PCR in these warning systems include the specific genes targeted, assumptions regarding how gene copy numbers correlate with toxin concentrations, and the exact method employed, whether quantitative, real time, or digital duplex PCR.

In a study by Ribeiro et al. [29], multiple primers were used for the detection of cylindrospermopsin (*cyrA*, *cyrB*, *cyrC*, and *cyrJ*), microcystins (*mcyE*), and saxitoxin (*sxtA*, *sxtB*, and *sxtI*). Samples were collected from the Billings Reservoir in São Paulo, Brazil, during a rainy and dry period. The *mcyE* gene was found in all samples, and, of the cylindrospermopsin genes tested, only *cyrB* and *cyrC* were found. The only sample positive for *cyrB* also tested positive for *cyrC.* Interestingly, for saxitoxin, some samples tested positive only for *sxtB* while other samples tested positive only for *sxtI*, and no samples were positive for *sxtA*.

From these results, Ribeiro et al. [29] concluded that using only one gene from a cyanotoxin gene cluster can lead to false negatives and false positives. False negatives can occur if there is a silent point mutation at the primer binding site and false positives can occur when other genes necessary for the biosynthesis of the targeted cyanotoxin are absent. Ribeiro et al. [29] note that other studies have found similar results where differential amplification between genes in the same gene cluster was observed.

Whatever gene clusters are used for PCR-based analysis of cyanotoxins, threshold values for corresponding gene copies to cyanotoxin concentrations need to be established. However, multiple factors can impact PCR results for cyanotoxins, such as primers potentially leading to false negatives or positives as mentioned previously, the assumption that gene copy numbers accurately represent toxicity, and other considerations more extensively covered by Pacheco et al. [30].

Given these complexities and potential sources of error, it is important to assess how reliably a given gene copy value corresponds to a cyanotoxin exceeding guideline values. Lu et al. (2019) [31] found values for gene copy numbers of *mcyB* (microcystin) and *pks* (cylindrospermopsin), which had a 10%, 25%, and 50% probability of corresponding to exceeding alert levels for microcystin and cylindrospermopsin. Lu et al. (2019) [31] tested a response-level model based on the 10% probability gene copy numbers and compared the results with ELISA. The level of agreement between qPCR and ELISA for the response level of thirty-six samples was calculated. The authors found excellent agreement between cyanotoxin gene copy numbers and ELISA results. They concluded that the proposed bio-marker model could be a reliable reference for response levels to toxic cyanobacteria blooms.

In another study, Lu et al. (2020) [32] developed an early warning system for toxic cyanobacteria blooms based on gene copy numbers. In this method, genes targeted general cyanobacteria, *Microcystis*, *Planktothrix*, *Cylindrospermopsis*, and *Nostoc*. Microcystin producers were also targeted by the genes *mcyEcya* and *mcyAcya*. This study found gene copy numbers corresponding to total microcystin levels of 0.3, 1.6, and 4 µg/L. Gene copy number values were determined using pooled ELISA data from four sites from Harsha Lake in Ohio, USA.

Lu et al. (2020) [32] tested both qPCR and reverse transcriptase (RT) qPCR. RT-qPCR was included to evaluate toxin gene expression in comparison with toxin levels. The authors found that *mcyAmic* determination by qPCR or RT-qPCR concentrations spiked approximately 3 weeks before LCMS or ELISA observed an increase in microcystins. These results show that qPCR could be valuable for monitoring the possibility of future harmful cyanobacteria blooms.

Zupančič et al. [33] collected planktonic water samples and biofilms mostly from rocks from multiple lakes and rivers in Slovenia. They compared qPCR testing for the *mcyE*, *cyrJ*, and *sxtA* genes with microscopic analysis and LC-MS/MS quantification of microcystins, cylindrospermopsin, and saxitoxin. Three *mcyE* assays, *cyrJ*, *sxtA*, and 16S cyano assays were tested.

The authors found non-target binding occurred for the 16S cyano assay and two microcystin assays. These results highlight the importance of the chosen qPCR assay for cyanotoxin detection. Ribeiro et al. [29] found some cyanotoxin genes may not be detected and lead to false negatives, and results by Zupančič et al. [33] show qPCRs potential for false negatives.

An important finding by Zupančič et al. [33] was the presence of cylindrospermopsin and saxitoxin genes in benthic mats while neither cyanotoxin was detected by LC-MS/MS. This means these water bodies have future cyanotoxin production potential, an important finding for future water management strategies.

### 3.4. Rapid Field Testing

A challenge with laboratory-based detection methods is that they can be time consuming and cost prohibitive. One alternative strategy for cheap, fast detection of cyanotoxins is the use of rapid test strips that can be used on site. In a study by Aranda-Rodriguez et al. [34], three field test kits for microcystin analysis were tested: the Abraxis strip test including QuickLyse^TM^, the Abraxis tube test, and the Envirologix tube test. This study looked at various factors such as specificity, sensitivity, and the rate of false positives or negatives and compared the results with ELISA and LCMS testing.

Aranda-Rodriguez et al. [34] found that the Abraxis strip test had a false negative rate of 2% and a false positive rate of 38%. The authors note that a false negative would be a greater public health concern as it would indicate no action is needed; however, the high false positive can have a cost impact if unnecessary actions are taken to further assess if cyanotoxins are present. They concluded that rapid test kits may be useful for screening for microcystins; however, different kits may not be useful depending on if they are quantitative, semi-quantitative, qualitative, and the concentration range they are meant to detect.

In another study, LeDuc et al. [35] assessed rapid test strips for the detection of microcystins, cylindrospermopsin, and anatoxin-a in comparison with ELISA results. The authors also looked at how tests were interpreted by trained vs. non-trained analysts. A difference of 14% for result interpretation was found between trained vs. non-trained analysts. One reason for this was that many samples had faint control lines that made interpretation more difficult. Overall, LeDuc et al. [35] concluded that although the strip test has potential for the rapid detection of cyanotoxins, improvements to their ease of use and interpretation would be beneficial.

Zvereva et al. [36] developed a silver-enhanced lateral flow immunoassay for the detection of microcystin-LR with increased sensitivity due to a hydroquinone-based reduction of silver ions on the surface of the gold nanoparticle. The LOD changed from 0.2 µg/L for a gold nanoparticle immunochromatographic assay to a LOD of 0.05 µg/L for the silver-enhanced assay. The test strip has a control line and test line which is dark at concentrations between 0 and 0.05 µg/L and mostly disappears by the time a concentration of 0.25 µg/L is reached. A test like this may increase the ease of interpretation of strip tests due to the lower LOD. However, this study used spiked bottled water, well water, and tap water samples. Further research using this method to test real-world samples would be needed.

Overall, test strips for cyanotoxin detection may be more cost effective and efficient than other more labor intense laboratory techniques. However, these tests would benefit from increased accuracy, precision, and ease of use. They would only be useful for early warning systems if used regularly for testing if other methods are not able to be employed due to cost.

### 3.5. Satellite Monitoring

Satellite monitoring has been used to identify algae blooms in many different bodies of water. Typically, satellite data is used to determine chlorophyll-a based on its absorption [37]. The WHO has released guidelines on how chlorophyll-a concentrations can be used as a proxy for cyanotoxin concentrations. Concentrations of chlorophyll-a below 24 µg/L are associated with no risk [38]. Many studies have used satellite data and various algorithms to identify toxic algal blooms.

One study by Mishra et al. [39] looked at the effectiveness of a cyanobacteria index (CI_cyano_) algorithm used to identify lakes with toxin-producing blooms. Mishra et al. [39] looked at multiple lakes across the United States over 11 bloom seasons. Satellite data was compared to measured microcystins in lake water using ELISA. It was found that the remote satellite monitoring had an 84% accuracy rate for detecting toxic blooms when compared with ELISA results. The authors conclude that satellite monitoring could be helpful for lake managers as a screening tool for where and when mitigation efforts should be focused. However, some limitations of the study include that the satellite does not detect cyanotoxins but uses intracellular pigments to evaluate cyanobacteria biomass. The authors also used microcystins as a proxy for all cyanotoxins and could have missed other toxins. Finally, lack of data for both bloom and non-bloom events has an impact. Testing is more likely to occur during harmful cyanobacteria blooms, which can bias models trained on available data.

Lopez Barreto et al. [38] evaluated satellite remote sensing chlorophyll-a measurements as a proxy for cyanobacteria in a major reservoir in California. Comparisons were made between satellite data and surface water samples collected by the California Department of Water Resources (DWR). The study found high agreement between cyanotoxin health advisories issued by the DWR and satellite remote sensing measurements of chlorophyll-a.

However, the authors note that satellite monitoring could miss nearly a quarter of public health advisories. This high false negative rate could be due to limitations including the spatial resolution of the satellite. Satellites can have a higher difficulty resolving nearshore points sampled by the DWR. Cloud cover was also noted as a possible hindrance to satellite measurements. Lopez Barreto et al. [38] note that satellite monitoring could be useful for informing when and where surface water samples should be taken. Satellite monitoring could also aid in issuing earlier warnings, especially in recreational areas while waiting for lab results.

### 3.6. Machine Learning Models

Machine learning could be a valuable tool for the early prediction of harmful algal blooms. These methods can combine multiple variables and trends over time to issue alerts. For example, Park et al. [40] used eleven different variables input into two different machine learning models to assess them for their ability to predict harmful cyanobacteria bloom alert levels in the Changnyung–Haman Reservoir in southeast Korea. Variables used included total dissolved nitrogen, total dissolved phosphorus, average air temperature, and average wind speed.

In another study, Gupta et al. [41] used machine learning models along with multiple variables to predict cyanobacteria blooms in Lake Erie 10, 20, and 30 days in advance. Satellite and in situ data were combined in the machine learning approach. Some variables included a cyanobacterial index (CI) calculated using Medium Resolution Imaging Spectroradiometer (MERIS) and Sentinel-3 OLCI (Ocean and Land Cover Instrument) satellite observations. Air temperature, total phosphorus, total Kjeldahl nitrogen, and total suspended solids data were also used. The authors concluded that their results were encouraging, but future work is needed to accommodate for uncertainties in the model. Computational costs and lack of available in situ data are also hurdles for these methods.

Kim et al. [42] optimized an algorithm for forecasting harmful cyanobacteria blooms after varying factors such as the data collection period used for training and the forecast lead time. The data were collected between 2013 and 2021 from the Nakdong River in South Korea. Variables included water quality data such as dissolved oxygen, total nitrogen, total phosphorous, water temperature, and chlorophyll-a. The best-performing model was trained data from a 7-year period with a 14-day lead forecasting time. The model was verified using data from unseen cyanobacteria bloom events.

Villanueva et al. [43] used data collected between 2018 and 2021 from a lake in Iowa to develop a predictive model for forecasting harmful cyanobacteria blooms one week in advance. Variables selected included *Microcystis mcyA* gene copies, pH, dissolved organic carbon, and orthophosphate. Three different models were trained and evaluated on multiple metrics such as accuracy, sensitivity, and specificity.

Machine learning models could be invaluable for early warning systems for harmful cyanobacteria blooms. However, they are limited by the need for large amounts of training data from previous years and, for the best results, a diverse range of variables should be included. Data can be collected from satellites and water samples including pH, total phosphorous, total nitrogen, and total suspended solids. The type of data and collection method could bring additional resource costs. For example, a machine learning model based solely on satellite data would require less continual effort than a model based on lab-gathered data.

## 4. Comparison of Methods for Early Warning Systems

Few studies directly compare multiple methods for their applicability in early warning systems for harmful cyanobacteria bloom detection. Table 2 summarizes results of papers which tested various methods for cyanotoxin detection and lists their pros and cons.

Immunoassays such as ELISA excel at being easily accessible, fast, and easy. However, they are not as sensitive as LCMS methods and can have issues due to cross-reactivity and matrix effects. LCMS methods are excellent for cross-comparing and validating other methods due to their sensitivity and specificity. However, LCMS methods may not be an ideal choice for early detection warning systems as they can have a high cost and would not be ideal for routine testing. Lastly, PCR-based methods have the potential to indicate the future risk of harmful cyanobacteria blooms before cyanotoxins are produced. However, few studies have attempted to correlate cyanotoxin gene copy numbers to cyanotoxin concentrations. More studies such as that by Lu et al. (2019) [31] would be needed to determine appropriate threshold values of gene copy numbers associated with a notable risk of cyanotoxin production.

## 5. Conclusions and Future Directions

Early warning systems for harmful cyanobacteria blooms are important for the health and safety of humans and animals. With the rise in harmful cyanobacteria blooms [8], effective monitoring is becoming increasingly important. As reviewed here, several methods for cyanobacteria toxin detection exist, each with pros and cons.

Tiered approaches have been suggested by Schürmann et al. [44] and Almuhtaram et al. [48] for cyanobacteria monitoring. Tier one would consist of rapid, cheaper, but less reliable methods compared with higher tiers. These may include methods such as microscopy and fluorometry to measure the biomass of a water body. These methods could be used on a regular basis since they are more accessible. A second-tier response would be prompted if tier one measures exceeded a pre-defined threshold value. Second-tier or third-tier measures would include more accurate and sensitive methods that have a greater cost or require more specialized equipment. These can include LC-MS/MS, qPCR, or immunoassays.

Another consideration for early warning systems is that many methods only detect the presence of cyanotoxins and not the possibility of future toxic events. However, detecting the presence of cyanotoxins at their onset is still valuable and early enough to mitigate adverse public health and economic impacts. Regardless, methods shown to have some capacity to predict future toxic algal blooms include PCR-based methods such as qPCR and machine learning models.

An example of qPCR prediction includes Duan et al. [49], who determined a threshold value for the *mcyG* microcystin encoding gene cluster, which indicated that the health advisory level for microcystins, 0.3 µg/L, would be exceeded in approximately one week. If implemented, this method could allow for preventative action to be taken. Another method discussed in this paper published by Lu et al. (2020) [32] used RT-qPCR to predict toxic cyanobacteria blooms and found that *mcyAmic* spiked three weeks before cyanotoxins were detected by LCMS or ELISA.

Machine learning models have also shown promise for forecasting toxic cyanobacteria blooms. Gupta et al. [41] used machine learning to predict toxic cyanobacteria blooms 10, 20, or 30 days in advance and Kim et al. [42] optimized a model for forecasting harmful blooms two weeks in advance. However, machine learning models require an immense amount of training data, making them computationally expensive and impractical depending on the resources available for data collection.

Overall, if resources are not limited it seems qPCR and machine learning models have the most promise for predicting future toxic cyanobacterial blooms. However, tiered systems involving methods such as fluorimetry, rapid testing, or other cheaper, accessible testing methods may be more feasible. Although these methods may not be able to predict future blooms, the ability to catch the onset of blooms before they pose a major public health risk is also valuable.

## Figures and Tables

**Table 1 pathogens-13-01047-t001:** Guidelines for anatoxins in drinking water.

Guidance Value for Drinking Water	Source
3.7 µg/L	Institut National de Sante’ Publique du Québec 2005 cited by Carrière et al. [19]
4 µg/L	California Environmental Protection Agency Office of Environmental Health Hazard Assessment [20]
30 µg/L (short term drinking water value)	WHO 2020 [18]

**Table 2 pathogens-13-01047-t002:** Comparison of methods for cyanotoxin detection.

Methods	Cyanotoxins	Pros	Cons	Reference
LC-MS/MS (LOD 0.001 µg/L)	12 MC congeners, NOD, ATX, and CYN	High sensitivity.	Lack of MC reference standards.	Hammoud et al. [24]
		High specificity.	Requires more advanced laboratory set-up.	
ELISA ^a^ (LOD 0.10 µg/L)	MCs	Possible to detect MC congeners not targeted by LCMS.	Matrix effects; Cross-reactivity of MCs; Possible overestimation due to degradation or transformation of products.	
		Relatively fast and easy.		
PPIA ^a^ (LOD 0.25 µg/L)	MCs	Possible to detect MC congeners not targeted by LCMS.	Lacks sensitivity.	
		Relatively fast and easy.	Possible over or underestimation of MCs.	
qPCR ^b^	MC, CYN, and STX	Samples with gene *mcyE* align with samples found to contain MCs in LCMS, ELISA, and PPIA.	Only used to confirm presence of cyanobacteria and genes. Not evaluated for use in monitoring programs.	
Fluorometry ^a^	Estimated chlorophyll-a associated with cyanobacteria	Cheaper and more easily accessible than methods such as LC-MS/MS.	Weak correlation with LC-MS/MS; Less accurate than direct measurements of cyanotoxins; Tended to overestimate risk to the public	Schürmann et al. [44]
Microscopy	Looked for the presence of 11 potentially toxic cyanobacteria	Cheaper and more easily accessible than methods such as LC-MS/MS.	Less accurate than direct measurements of cyanotoxins.	
			Tended to overestimate risk to the public.	
qPCR ^b^	16s, *mcyE*	Strong correlation between *mcyE* qPCR and LC-MS/MS.	16S qPCR had a weak correlation with LC-MS/MS; Some samples with high MC levels had low *mcyE* copy numbers; No current guideline values for qPCR risk assessment.	
ELISA ^a^	MCs and ATX	Strong correlation with LC-MS/MS.	Overestimated MCs in comparison with LC-MS/MS.	
LC-MS/MS ^a^	MCs, ATX, CYN,	Assumed to be the “gold standard” for cyanotoxin detection.	More expensive.	
			Longer sample processing times.	
Shotgun metagenomics ^b^	Characterization of multiple cyanobacteria genera and gene abundance	Bloom vs. non-bloom samples could be differentiated by the abundance of various genes such as those for nitrogen and phosphorus metabolism.	Cyanotoxin genes not detected.	Saleem et al. [45]
			Not as effective as ELISA and qPCR for real-time toxigenic potential analysis.	
qPCR ^b^	16S RNA, *mycE*, *stxA*	Relatively fast compared with shotgun sequencing; *mcyE* gene copies correlated with MC/NOD levels; Better for real-time toxigenic potential analysis than shotgun sequencing.	Purified DNA found fewer gene copies for cyanotoxins than crude DNA. Purified DNA may underrepresent cyanotoxin potential while crude DNA has more PCR inhibition.	
			Does not provide as much information on the cyanobacteria community present as shot-gun sequencing.	
ELISA ^a^	MCs/NODs	Relatively fast compared with shotgun sequencing.	Does not provide as much information on the cyanobacteria community present as shot-gun sequencing.	
		Better for real-time toxigenic potential analysis than shotgun sequencing.		
LC-MS/MS ^a^	MCs, NOD, CYN, ATX	Analysis of microbial mats possible.	Unable to determine an MC structure in one sample based on fragments.	Khomutovska et al. [46]
		More reliable than ELISA.		
qPCR ^b^	*mcyA*, *mcyD*, *mcyE*, *mcyE/ndaF*, *sxtA*, and *anaC*	Analysis of microbial mats possible.	Primers may not be universal for both planktonic and benthic mat cyanobacteria.	
		More information on the microbial community present.		
ELISA ^a^	MCs (LOD 0.5 ng/L), CYN (LOD 0.5 ng/L), ATX (LOD 0.4 ng/L)	Rapid, results in ~30 min.	ATX was found in six samples, but not by LC-MS/MS or qPCR. False positives.	
Mass spectrometry multiple reaction monitoring (MS-MRM) ^a^	MCs, STX, CYN	More sensitive than ELISA.	Same samples where MCs were detected by ELISA and *mcyE* genes were detected by qPCR were not positive for MCs by MS-MRM.	McKindles et al. [47]
qPCR ^b^	*mcyE*, *stxA*, and *cyrA*	Potential as a valuable early warning tool, especially in the summer.	Some samples which had MCs or CYN did not have detectable *mcyE* or *cyr* genes.	
ELISA ^a^	MCs		Lower sensitivity compared with HPLC methods.	

^a^ indicates methods used for the detection of low concentrations of cyanotoxins, ^b^ indicates methods with the potential to predict future cyanotoxin production.

## Data Availability

No new data were created or analyzed in this study.

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
