# Peer review of "Early Detection Methods for Toxic Cyanobacteria Blooms"

_pathogens, 2024, doi:10.3390/pathogens13121047_

Round 1
Reviewer 1 Report
Comments and Suggestions for Authors
This review provides a thorough comparison of cyanobacterial detection methods, but it lacks practical application examples and discussion on the cost-effectiveness of advanced techniques. Including more real-world case studies on how these methods are implemented in monitoring systems would help readers better understand their practical utility. Furthermore, while PCR methods are highlighted as promising tools, the paper does not sufficiently address the limitations in setting gene copy thresholds and their correlation with toxin concentrations. Additional research and experimental data are necessary to support the reliability of this method. Finally, there is inadequate discussion on the cost and feasibility of advanced methods such as LC-MS, particularly in resource-limited areas. Exploring alternative, cost-effective methods would enhance the study’s practical relevance and improve its impact in real-world applications. Please refer to the attached file for detailed comments.

Author Response
Comments and Suggestions for Authors:
This review provides a thorough comparison of cyanobacterial detection methods, but it lacks practical application examples and discussion on the cost-effectiveness of advanced techniques. Including more real-world case studies on how these methods are implemented in monitoring systems would help readers better understand their practical utility. Furthermore, while PCR methods are highlighted as promising tools, the paper does not sufficiently address the limitations in setting gene copy thresholds and their correlation with toxin concentrations. Additional research and experimental data are necessary to support the reliability of this method. Finally, there is inadequate discussion on the cost and feasibility of advanced methods such as LC-MS, particularly in resource-limited areas. Exploring alternative, cost-effective methods would enhance the study’s practical relevance and improve its impact in real-world applications. Please refer to the attached file for detailed comments.
Response: We thank reviewer-1 for recognizing that this work will expand the knowledge on the cyanobacterial detection methods and can contribute to increase the awareness of harmful algal blooms. A completely new section alternative, cost-effective “Rapid Field Testing” has been included to enhance the study’s practical relevance and improve its impact in real-world applications. We have included new studies such satellite monitoring, and, more recently, machine learning model that may be able to predict future blooms i.e. the ability to catch the onset of blooms before they pose a major public health risk. Please refer to the line 276 to 380 of corresponding revisions/corrections highlighted/in track changes in the re-submitted files. The pdf file where reviewer-1 highlighted few section for corrections is attached with appropriate response and these corrections has been addressed in the submitted revised version as well.

Reviewer 2 Report
Comments and Suggestions for Authors
Pathogens_Grant et al. 2024
This manuscript addresses an important need—how to identify harmful algal blooms before they actually bloom. If a successful method could be developed, that provides sufficient warning time, and is both scalable and transferable, it would have a profound effect on management of water bodies.
Major Comments:
The paper is a nice review of methods to detect microcystins but it does not address the issue stated in the title: “early warning system”. In most of the examples provided, these methods are being used to detect cyanotoxins at the onset or after the bloom, not before it starts. If the authors wish to retain the current structure, a more accurate title for this paper would be “Detection Methods for Toxicity of Cyanobacterial Blooms.” If they wish to emphasize the early detection theme, the paper needs to be restructured to focus on true early detection so that lake managers are proactive, not reactive.
Specific Comments (page/line):
69: I don’t think “neglected” is the appropriate word here; perhaps replace “neglected the” with “does not”. More importantly, it would be very helpful if the absolute difference in microcystin concentration between the two methods was included, and not just one captured them and another did not.
96: This concentration would have little meaning to someone not working in this field; is it possible to provide context to the major cyanotoxin concentrations by providing a table that lists the main cyanotoxins, the concentrations ranges observed based on your citations, and the MECs for each toxin? This would help place the concentrations listed in each method in some context.
211-216: The study discussed in this paragraph is really the only true example of an “early warning system”, whereby the methods result in information that can be used to predict whether a bloom is likely (or not) to be imminent. The other studies, at least as currently written, describe the methods to detect potential blooms (or toxicity) but do not link to predictive ability. This point is acknowledged on lines 241-245 although not elaborated on.
Table 1: I recommend an addition column to this table be added that indicates whether the method is suitable for an early warning system or simply for detection at low concentrations, and if so, the minimum level of detection for the analyte under consideration.
Editorial Comments: (line):
- 13: replace “included” with “include” as this should be present tense to be consistent with “current”
- 109: why the hyphen between chromatography and mass when it is not used elsewhere? I am fine with a hyphen (or slash) between the two words (or not), but whatever option they choose needs to be applied consistently throughout the manuscript.
- 206: is the “I” after Microcystis intentional (to identify a certain genotype?) or a typo?
Author Response
This manuscript addresses an important need—how to identify harmful algal blooms before they actually bloom. If a successful method could be developed, that provides sufficient warning time, and is both scalable and transferable, it would have a profound effect on management of water bodies.
We thank reviewer-2 for taking the time to review this manuscript. Please find the detailed responses below and the corresponding revisions/corrections highlighted/in track changes in the re-submitted file.
Major Comments:
The paper is a nice review of methods to detect microcystins but it does not address the issue stated in the title: “early warning system”. In most of the examples provided, these methods are being used to detect cyanotoxins at the onset or after the bloom, not before it starts. If the authors wish to retain the current structure, a more accurate title for this paper would be “Detection Methods for Toxicity of Cyanobacterial Blooms.” If they wish to emphasize the early detection theme, the paper needs to be restructured to focus on true early detection so that lake managers are proactive, not reactive.
Response: We agree with the reviewer the earlier version lacked studies except qPCR mostly used in early detection warning systems. As suggested by the reviewer, we have included new studies such satellite monitoring, and, more recently, machine learning model that does address the issue stated in the title and may be able to predict future blooms i.e. the ability to catch the onset of blooms before they pose a major public health risk. Therefore, we prefer to keep the same title. Please refer to the line 315 to 380 of corresponding revisions/corrections highlighted/in track changes in the re-submitted file.
Specific Comments (page/line):
69: I don’t think “neglected” is the appropriate word here; perhaps replace “neglected the” with “does not”. More importantly, it would be very helpful if the absolute difference in microcystin concentration between the two methods was included, and not just one captured them and another did not.
Response: “neglected” changed with “does not”. Please see Line 89 tracked changes File
96: This concentration would have little meaning to someone not working in this field; is it possible to provide context to the major cyanotoxin concentrations by providing a table that lists the main cyanotoxins, the concentrations ranges observed based on your citations, and the MECs for each toxin? This would help place the concentrations listed in each method in some context.
Response: A table that lists the main cyanotoxins have been provided. Please see Line 134 tracked changes File
211-216: The study discussed in this paragraph is really the only true example of an “early warning system”, whereby the methods result in information that can be used to predict whether a bloom is likely (or not) to be imminent. The other studies, at least as currently written, describe the methods to detect potential blooms (or toxicity) but do not link to predictive ability. This point is acknowledged on lines 241-245 although not elaborated on.
Response: This point has been elaborated further with inclusion of additional information relevant to this comment.
Table 1: I recommend an addition column to this table be added that indicates whether the method is suitable for an early warning system or simply for detection at low concentrations, and if so, the minimum level of detection for the analyte under consideration.
Response: The information whether the method is suitable for an early warning system or simply for detection has been provided as foot note in the table.2. Please see Line 400-401 tracked changes File
Editorial Comments: (line):
- 13: replace “included” with “include” as this should be present tense to be consistent with “current”
Response: Change has been applied Please see Line 13 tracked changes File
- 109: why the hyphen between chromatography and mass when it is not used elsewhere? I am fine with a hyphen (or slash) between the two words (or not), but whatever option they choose needs to be applied consistently throughout the manuscript.
Response: Change has been applied as per reviewers’ suggestion throughout the manuscript
- 206: is the “I” after Microcystis intentional (to identify a certain genotype?) or a typo?
Response: The typo ”I” has been corrected.
Round 2
Reviewer 2 Report
Comments and Suggestions for Authors
The authors have done a nice job overall in responding to my prior concerns; unfortunately, I still have a few comments on the revised version:
1) while I won't fall on my sword over this, I still don't believe this paper addresses "early warning systems". Early warning systems would include much more than just an early analysis; it would include the protocols involved in warning the public, which is not a trivial consideration. What this manuscript does address is "early detection systems" and if I was the managing editor, I would insist on the title being changed accordingly.
2) I assume Fig. 1 was added in response to a different reviewer's comments. Frankly, I don't think it adds much to the manuscript, and indeed may confuse rather than illuminate. I simply don't see what value it adds to the paper.
3) line 319: This sentence is too strong and the caveat provided in the next paragraph should be located immediately after the sentence, not multiple afterwards.
Author Response
Author's Response to the Review Report (Reviewer 2), Round2
Comments and Suggestions for Authors
Pathogens_Grant et al. 2024
The authors have done a nice job overall in responding to my prior concerns; unfortunately, I still have a few comments on the revised version:
We thank reviewer-2 for taking the time to review this manuscript.
1) while I won't fall on my sword over this, I still don't believe this paper addresses "early warning systems". Early warning systems would include much more than just an early analysis; it would include the protocols involved in warning the public, which is not a trivial consideration. What this manuscript does address is "early detection systems" and if I was the managing editor, I would insist on the title being changed accordingly.
Response: The title has been changed to “Early detection Methods for Toxic Cyanobacteria Blooms”
2) I assume Fig. 1 was added in response to a different reviewer's comments. Frankly, I don't think it adds much to the manuscript, and indeed may confuse rather than illuminate. I simply don't see what value it adds to the paper.
Response: Figure 1 has been removed
3) line 319: This sentence is too strong, and the caveat provided in the next paragraph should be located immediately after the sentence, not multiple afterwards.
Response: The sentence has been revised.